# LLM-ESR: Large Language Models Enhancement for Long-tailed Sequential Recommendation

**Qidong Liu[1,2], Xian Wu[3] [*], Yejing Wang[2], Zijian Zhang[2, 4],**
**Feng Tian[5] [*], Yefeng Zheng[3, 6], Xiangyu Zhao[2] [*]**
[1] School of Auto. Science & Engineering, MOEKLINNS Lab, Xi'an Jiaotong University
[2] City University of Hong Kong
[3] Jarvis Research Center, Tencent YouTu Lab, [4] Jilin University
[5] School of Comp. Science & Technology, MOEKLINNS Lab, Xi'an Jiaotong University
[6] Medical Artificial Intelligence Lab, Westlake University
`liuqidong@stu.xjtu.edu.cn`, {`kevinxwu, yefengzheng`}`@tencent.com`,
`yejing.wang@my.cityu.edu.hk`, `zhangzijian@jlu.edu.cn`,
`fengtian@mail.xjtu.edu.cn`, `xianzhao@cityu.edu.hk`

## Abstract

Sequential recommender systems (SRS) aim to predict users' subsequent choices based on their historical interactions and have found applications in diverse fields such as e-commerce and social media. However, in real-world systems, most users interact with only a handful of items, while the majority of items are seldom consumed. These two issues, known as the long-tail user and long-tail item challenges, often pose difficulties for existing SRS. These challenges can adversely affect user experience and seller benefits, making them crucial to address. Though a few works have addressed the challenges, they still struggle with the seesaw or noisy issues due to the intrinsic scarcity of interactions. The advancements in large language models (LLMs) present a promising solution to these problems from a semantic perspective. As one of the pioneers in this field, we propose the **L**arge **L**anguage **M**odels **E**nhancement framework for **S**equential **R**ecommendation (**LLM-ESR**). This framework utilizes semantic embeddings derived from LLMs to enhance SRS without adding extra inference load from LLMs. To address the long-tail item challenge, we design a dual-view modeling framework that combines semantics from LLMs and collaborative signals from conventional SRS. For the long-tail user challenge, we propose a retrieval augmented self-distillation method to enhance user preference representation using more informative interactions from similar users. To verify the effectiveness and versatility of our proposed enhancement framework, we conduct extensive experiments on three real-world datasets using three popular SRS models. The results show that our method surpasses existing baselines consistently, and benefits long-tail users and items especially. The implementation code is available at https://github.com/Applied-Machine-Learning-Lab/LLM-ESR.

## 1 Introduction

The objective of sequential recommendation is to predict the next likely item for users based on their historical records [7, 54]. Owing to its wide-ranging applicability in various domains such as e-commerce [48] and social media [5], sequential recommendation has garnered considerable attention in recent years. Given that the essence of sequential recommendation revolves around extracting user preferences from their interaction records, several innovative architectures have been proposed. For

---

[*]Corresponding authors: Xian Wu, Feng Tian and Xiangyu Zhao

38th Conference on Neural Information Processing Systems (NeurIPS 2024).

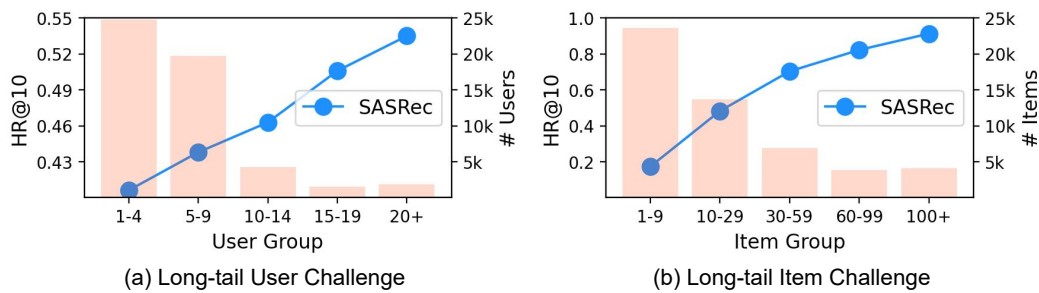

(a) Long-tail User Challenge

(b) Long-tail Item Challenge

Figure 1: The preliminary experiments of SASRec on Beauty dataset.

instance, SASRec [18] applies the self-attention technique to capture the users' long-term preference, while FMLPRec [24] introduces a pure MLP architecture to identify dynamics in users' preference.

Despite significant advancements in sequential recommendation, the long-tail challenges continue to undermine its practical utility. Generally, these challenges can be categorized into two types, affecting either the user or the item side. To illustrate, we present the performance of a well-known SRS model, SASRec [18], on the Amazon Beauty dataset, along with its statistics in Figure 1. **i) Long-tail User Challenge**: In Figure 1 (a), we note that above $80\%$ users have interacted with fewer than 10 items (*i.e.,* long-tail users), and SASRec's performance is subpar for these users compared to those with more interaction records. This suggests that the majority of users receive less than optimal recommendation services. **ii) Long-tail Item Challenge**: Figure 1 (b) demonstrates that SASRec performs significantly better on more popular items. However, the histogram indicates that around $71.4\%$ items own no more than 30 interaction records, meaning they are less frequently consumed. Addressing these long-tail challenges is crucial for elevating user experience and seller benefits.

To tackle the long-tail item challenge, existing studies [17, 20] examine the co-occurrence pattern between popular and long-tail items, aiming to enrich the representation of long-tail items with that of popular ones. Nevertheless, ignorance of the true relationship between items may cause a seesaw problem [36]. As for the long-tail user challenge, existing research [37, 34] explores the interaction history of all users, attempting to augment pseudo items for tail users. However, these approaches still only rely on collaborative information, which inclines to generate noisy items due to inaccurate similarity between users [34]. At this time, superb semantic relations between users or items can make an effect, which indicates the potential of utilizing semantics to face long-tail challenges.

Recent advancements in large language models (LLMs) offer promise for alleviating long-tail challenges from a semantic perspective. However, LLMs are initially designed for natural language processing tasks but not for recommendation ones. Some works [63, 43] have made efforts to adapt, but two problems still exist. **i) Inefficient Integration**: Recent research has explored deriving informative prompts to activate ChatGPT [55, 10] or modifying the tokenization method of LLaMA [25, 27, 59] for sequential recommendation. Despite their impressive performance, these approaches are challenging to apply in industrial settings. This is because recommender systems typically require low latency for online deployment, whereas LLMs often entail high inference costs [11]. **ii) Deficiency of Semantic Information**: Several recent works [13, 16] propose utilizing embeddings derived from LLMs to initialize the item embedding layer of sequential recommendation models, thereby integrating semantic information. However, the fine-tuning process, if not done without freezing the embedding layer, may erode the original semantic relationships between items. Additionally, these approaches focus solely on the item side, neglecting the potential benefits of incorporating semantic information on the user side which could aid the sequence encoder of an SRS.

In this paper, to better integrate LLMs into SRS for addressing long-tail challenges, we design a **L**arge **L**angauge **M**odels **E**nhancement framework for **S**equential **R**ecommendation (**LLM-ESR**). Firstly, we derive the semantic embeddings of items and users by encoding prompt texts from LLMs. Since these embeddings can be cached in advance, our integration does not impose any extra inference burden from LLMs. To tackle the long-tail item challenge, we devise a dual-view modeling framework that combines semantic and collaborative information. Specifically, the embeddings derived from LLMs are frozen to avoid deficiency of semantics. Next, we propose a retrieval augmented self-distillation method to enhance the sequence encoder of an SRS model using similar users. The similarity between users is measured by the user representations from LLMs. Finally, it is important

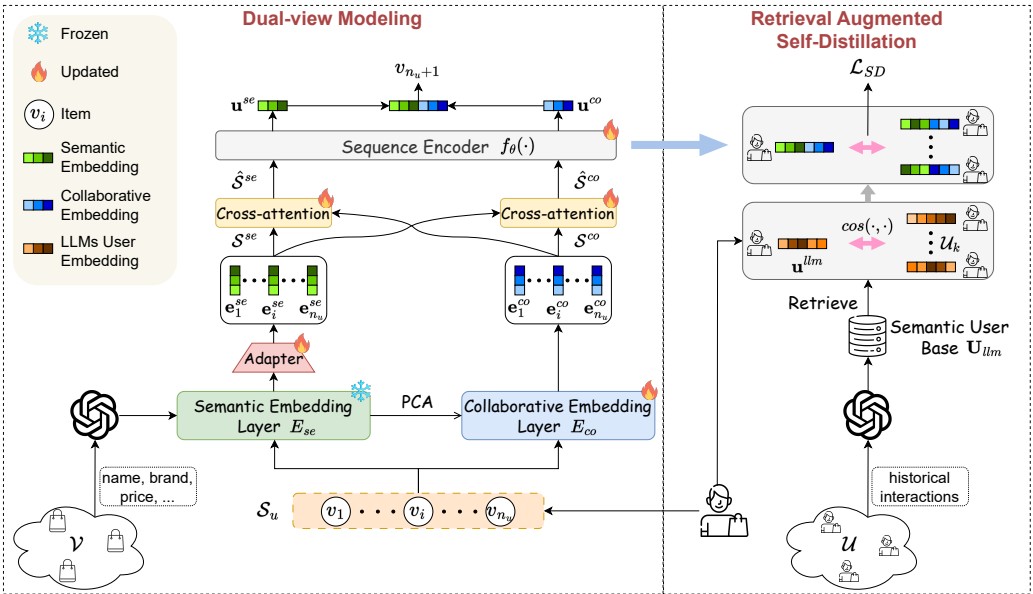

Figure 2: The overview of the proposed LLM-ESR framework.

to note that the proposed framework is model-agnostic, allowing it to be adapted to any sequential recommendation model. The contributions of this paper are as follows:

- We propose a large language models enhancement framework, which can alleviate both long-tail user and item challenges for SRS by introducing semantic information from LLMs.

- To avoid the inference burden of LLMs, we design an embedding-based enhancement method. Besides, the derived embeddings are utilized directly to retain the original semantic relations.

- We conduct extensive experiments on three real-world datasets with three backbone SRS models to validate the effectiveness and flexibility of LLM-ESR.

## 2 Problem Definition

The goal of the sequential recommendation is to give out the next item that users are possible to interact with based on their interaction records. The set of users and items are denoted as $\mathcal{U} = \{u_1, \ldots, u_i, \ldots, u_{|\mathcal{U}|}\}$ and $\mathcal{V} = \{v_1, \ldots, v_i, \ldots, v_{|\mathcal{V}|}\}$, respectively, where $|\mathcal{U}|$ and $|\mathcal{V}|$ are the number of users and items. Each user has an interaction sequence, which arranges the interacted items by timeline, denoted as $\mathcal{S}_u = \{v_1^{(u)}, \ldots, v_i^{(u)}, \ldots, v_{n_u}^{(u)}\}$. $n_u$ represents the interaction number of user $u$. For simplicity, we omit the superscript $(u)$ in the following sections. Then, the problem of sequential recommendation can be defined as follows:

$$arg \max_{v_i \in \mathcal{V}} P(v_{n_u+1} = v_i | \mathcal{S}_u) \tag{1}$$

Following the existing works related to long-tailed SRS [17, 20], we can split the users and items into tail and head groups. Let $n_u$ and $p_v$ denote the length of the user's interaction sequence and the popularity of the item $v$ (*i.e.,* the total interaction number). Firstly, we sort the users and items by the values of $n_u$ and $p_v$ in descending order. Then, take out the top $20\%$ users and items as **head user** and **head item** according to Pareto principle [4], denoted as $\mathcal{U}_{head}$ and $\mathcal{V}_{head}$. The rest of the users and items are the **tail user** and **tail item**, *i.e.,* $\mathcal{U}_{tail} = \mathcal{U} \setminus \mathcal{U}_{head}$ and $\mathcal{V}_{tail} = \mathcal{V} \setminus \mathcal{V}_{head}$. To alleviate the long-tail challenges, we aim to elevate the recommending performance for $\mathcal{U}_{tail}$ and $\mathcal{V}_{tail}$.

## 3 LLM-ESR

### 3.1 Overview

The overview of the proposed LLM-ESR is shown in Figure 2. To acquire the semantic information, we adopt LLMs to encode textual users' historical interactions and items' attributes into LLMs user embedding and LLMs item embedding. Then, two modules are proposed to augment long-tail items and long-tail users, respectively, *i.e.,* Dual-view Modeling and Retrieval Augmented Self-Distillation. **i) Dual-view Modeling**: This module consists of two branches. One is *semantic-view modeling*, which aims to extract the semantic information from the user's interaction sequence. It first utilizes the semantic embedding layer, derived from LLMs item embedding, to encode the items. Then, an adapter is designed for dimension adaptation and space transformation. The output item embedding sequence will be fed into cross-attention for fusion and then sequence encoder to get the user representation in semantic view. The other branch is *collaborative-view modeling*, which transforms the interaction sequence into an embedding one by a collaborative embedding layer. Next, followed by a cross-attention and the sequence encoder, the collaborative user preference is obtained. At the end of this module, the user representations in the two views will be fused for the final recommendations. **ii) Retrieval Augmented Self-Distillation**: This module expects to enhance long-tail users through informative interactions of similar users. First, the derived LLMs user embedding is considered as a semantic user base for retrieving similar users. Then, similar users are fed into dual-view modeling to get their user representations, which are the guide signal for self-distillation. Finally, the derived distillation loss will be utilized as an auxiliary loss for training.

### 3.2 Dual-view Modeling

The traditional SRS models are skilled in capturing collaborative signals, which can recommend for popular items well [20, 17]. However, they compromise on long-tail items due to the lack of semantics [2]. Therefore, we model the preferences of users from the dual views to cover all items simultaneously. Besides, we propose a two-level fusion to better combine the benefits from both two.

**Semantic-view Modeling**. In general, the attributes and descriptions of items contain abundant semantics. To utilize the powerful semantic understanding abilities of LLMs, we organize the attributes and descriptions into textual prompts (the template of prompts can be found in **Appendix** A.1). Then, in avoid of possible inference burden brought by LLMs, we cache the embeddings derived from LLMs for usage. In specific, the embeddings can be obtained by taking out the last hidden state of open-sourced LLMs, such as LLaMA [51], or the public API, such as text-embedding-ada-002[2]. We adopt the latter one in this paper. Let $\mathbf{E}_{se} \in \mathbb{R}^{|\mathcal{V}| \times d_{llm}}$ denotes the LLMs embedding of all items, where $d_{llm}$ is dimension of LLMs embedding. Then, the semantic embedding layer $\mathbf{E}_{se}$ from LLMs can be used for semantic-view modeling to enhance long-tail items. However, previous works [13, 16] often adapt it as the initialization of the item embedding layer, which may ruin the original semantic relations during fine-tuning. In order to retain the semantics, we freeze the $\mathbf{E}_{se}$ and propose an adapter to transform the raw semantic space into the recommending space. For each item $i$, we can get its LLMs embedding $\mathbf{e}_i^{llm}$ by taking the $i$-th row of $\mathbf{E}_{se}$. Then, it will be fed into the tunable adapter to get the semantic embedding:

$$\mathbf{e}_i^{se} = \mathbf{W}_2^a(\mathbf{W}_1^a\mathbf{e}_i^{llm} + \mathbf{b}_1^a) + \mathbf{b}_2^a \qquad (2)$$

where $\mathbf{W}_1^a \in \mathbb{R}^{\frac{d_{llm}}{2} \times d_{llm}}$, $\mathbf{W}_2^a \in \mathbb{R}^{d \times \frac{d_{llm}}{2}}$ and $\mathbf{b}_1^a \in \mathbb{R}^{\frac{d_{llm}}{2} \times 1}$, $\mathbf{b}_2^a \in \mathbb{R}^{d \times 1}$ are the weight matrices and bias of adapter. Following this process, we can obtain the item embedding sequence of the user's interaction records, denoted as $\mathcal{S}^{se} = [\mathbf{e}_1^{se}, \dots, \mathbf{e}_{n_u}^{se}]$. Similar to a general SRS model, we employ a sequence encoder $f_\theta$ (*e.g.,* self-attention layers [52] for SASRec [18]) to get the representation of user preference in semantic view as follows:

$$\mathbf{u}^{se} = f_\theta(\mathcal{S}^{se}) \qquad (3)$$

where $\mathbf{u}^{se} \in \mathbb{R}^{d \times 1}$ is the user preference representation in semantic view and $\theta$ denotes the parameters of sequence encoder in an SRS model.

**Collaborative-view Modeling**. To utilize the collaborative information, we adopt a trainable item embedding layer and supervised update it by interaction data. Let $\mathbf{E}_{co} \in \mathbb{R}^{|\mathcal{V}| \times d}$ denotes the

---

[2]https://platform.openai.com/docs/guides/embeddings

collaborative embedding layer of the item. Then, the item embedding sequence $\mathcal{S}^{co} = [\mathbf{e}_1^{co}, \ldots, \mathbf{e}_{n_u}^{co}]$ is acquired by extracting the corresponding rows from $\mathbf{E}_{co}$. To get the user preference $\mathbf{u}^{co}$ in the collaborative view, we input embedding sequence to sequence encoder, i.e., $\mathbf{u}^{co} = f_\theta(\mathcal{S}^{co})$. It is worth noting that, the sequence encoder $f_\theta$ is the same one in both semantic and collaborative views for the shared sequential pattern and higher efficiency [46]. Besides, the embedding layers in the two views are in unbalanced training stages (one is pretrained, while the other is from scratch), which may lead to optimization difficulty [1]. To handle such a problem, we initialize the $\mathbf{E}_{co}$ by dimension-reduced $\mathbf{E}_{se}$. The Principal Component Analysis (PCA) [44] is used as the dimension reduction method in this paper.

**Two-level Fusion**. The effective integration of both semantic-view and collaborative-view is essential to absorb the benefits of these two. However, the direct merge of the user representations in dual views may overlook the nuanced inter-relationships between item sequences. Thus, we design a two-level fusion method for the dual-view modeling module, i.e., sequence-level and logit-level. The former aims to implicitly capture the mutual relationships between the item sequences of dual views, while the latter explicitly targets the combination of recommending abilities. In specific, we propose a cross-attention mechanism for sequence-level fusion. To simplify the description, we only take the semantic view interacting with the collaborative view for illustration, and the other view is the same. Specifically, $\mathcal{S}^{se}$ is considered as the *query*, and $\mathcal{S}^{co}$ as the *key* and *value* in attention mechanism. Let $\mathbf{Q} = \mathcal{S}^{se}\mathbf{W}^Q, \mathbf{K} = \mathcal{S}^{co}\mathbf{W}^K, \mathbf{V} = \mathcal{S}^{co}\mathbf{W}^V$, where $\mathbf{W}^Q, \mathbf{W}^K, \mathbf{W}^V \in \mathbb{R}^{d \times d}$ are weight matrices. Then, the interacted collaborative embedding sequence can be formulated as follows:

$$\hat{\mathcal{S}}^{co} = \text{Softmax}(\frac{\mathbf{Q}\mathbf{K}^T}{\sqrt{d}})\mathbf{V} \tag{4}$$

Following the same process of cross-attention, we can also get the corresponding semantic embedding sequence $\hat{\mathcal{S}}^{se}$. Finally, $\mathcal{S}^{se}, \mathcal{S}^{co}$ are substituted by $\hat{\mathcal{S}}^{se}, \hat{\mathcal{S}}^{se}$ to be fed into $f_\theta(\cdot)$. As for logit-level fusion, we concatenate the two-view user and item embeddings for recommendation. The probability score of recommending item $j$ for the user $u$ is therefore calculated as:

$$P(v_{n_u+1} = v_j | v_{1:n_u}) = [\mathbf{e}_j^{se} : \mathbf{e}_j^{co}]^T [\mathbf{u}^{se} : \mathbf{u}^{co}] \tag{5}$$

where ":" denotes the concatenation operation of two vectors. Based on the probability score, we adopt the pairwise ranking loss to train the framework:

$$\mathcal{L}_{Rank} = -\sum_{u \in \mathcal{U}} \sum_{k=1}^{n_u} \log\sigma(P(v_{k+1}^+ = |v_{1:k}) - P(v_{k+1}^- = |v_{1:k})) \tag{6}$$

where $v_{k+1}^+$ and $v_{k+1}^-$ are the ground-truth item and paired negative item. It is worth noting that the ranking loss may differ a little according to different backbone SRS models, e.g., sequence-to-one pairwise loss for GRU4Rec [14].

### 3.3 Retrieval Augmented Self-Distillation

The long-tail user problem originates from the lack of enough interactions for the sequence encoder in an SRS to capture users' preferences. Thus, we propose a self-distillation method to augment the extraction capacity of the sequence encoder. Self-distillation [12, 61] is a type of knowledge distillation that considers one model as both the student and teacher for model enhancement. As for the SRS, since multiple similar users have more informative interactions, it is promising to transfer their knowledge to the target user for strengthening. Thereafter, there are two key challenges for such knowledge transfer, i.e., how to retrieve similar users and how to transfer the knowledge.

**Retrieve Similar Users**. Previous works have confirmed that LLMs can understand the semantic meanings of textual user interaction records for recommendation [25, 10]. Based on their observation, we organize the item's title that interacted by users into the textual prompts (the template of prompts can be found in **Appendix** A.1). Then, similar to the derivation of LLMs item embedding $\mathbf{E}_{se}$, we can obtain and save the LLMs user embedding, denoted as $\mathbf{U}_{llm} \in \mathbb{R}^{|\mathcal{U}| \times d_{llm}}$. It is also dubbed as the semantic user base in this paper, because the semantic relations are encoded in it. For each target user $k$, we can retrieve the similar user set $\mathcal{U}_k$ as follows:

$$\mathcal{U}_k = \text{Top}(\{\cos(\mathbf{u}_k^{llm}, \mathbf{u}_j^{llm})\}_{j=1}^{|\mathcal{U}|}, N) \tag{7}$$

where $\cos(\cdot, \cdot)$ is the cosine similarity function to measure the distance between two vectors. $N$ represents the size of similar user sets, which is a hyper-parameter.

**Self-Distillation**. As mentioned before, we design the self-distillation to transfer the knowledge from several similar users to the target user. Since the representation of user preference, *i.e.,* $\mathbf{u}^{se}$ and $\mathbf{u}^{co}$, encode the comprehensive knowledge of the user, we configure such representation as the mediator for the distillation. To get the teacher mediator, we first utilize the dual-view modeling framework (Section 3.2) to get the user representation for each similar user, denoted as $\{\mathbf{u}_j^{se}, \mathbf{u}_j^{co}\}_{j=1}^{|\mathcal{U}_k|}$. Then, the teacher mediator is calculated by mean pooling, as the following formula:

$$[\mathbf{u}_{T_k}^{se} : \mathbf{u}_{T_k}^{co}] = \text{Mean\_Pooling}(\{[\mathbf{u}_j^{se} : \mathbf{u}_j^{co}]\}_{j=1}^{|\mathcal{U}_k|}) \tag{8}$$

The student mediator is the representation of target user $k$, *i.e.,* $[\mathbf{u}_k^{se} : \mathbf{u}_k^{co}]$. Based on the teacher and student mediators, the self-distillation loss can be formulated as:

$$\mathcal{L}_{SD} = \frac{1}{|\mathcal{U}|} \sum_{k=1}^{|\mathcal{U}|} |[\mathbf{u}_k^{se} : \mathbf{u}_k^{co}] - [\mathbf{u}_{T_k}^{se} : \mathbf{u}_{T_k}^{co}]|^2 \tag{9}$$

Note that the gradients of $\mathbf{u}_{T_k}^{se}$ and $\mathbf{u}_{T_k}^{co}$ are stopped, because they only provide the guidance signal instead of optimizing the model.

### 3.4 Train and Inference

**Train**. Based on the illustration in Section 3.2 and Section 3.3, we only update the collaborative embedding layer, adapter, cross-attention and sequence encoder during the training, while freezing the semantic embedding layer and semantic user base. Since the original LLMs embeddings $\mathbf{E}_{se}$ and $\mathbf{U}_{llm}$ are frozen, the original semantic relations get preserved well. The training loss for optimization is the combination of pairwise ranking loss and self-distillation loss, which can be written as follows:

$$\mathcal{L} = \mathcal{L}_{Rank} + \alpha \cdot \mathcal{L}_{SD} \tag{10}$$

where $\alpha$ is a hyper-parameter to adjust the magnitude of self-distillation.

**Inference**. During the inference process of the LLM-ESR, the retrieval augmented self-distillation module is exempted due to no need for the auxiliary loss. Thus, we follow the dual-view modeling process for the final recommendation by Equation (5). Besides, since the semantic embedding layer can be cached in advance, the call for LLMs is avoided, which prevents the extra inference costs. Due to the limited space, the algorithm lies in **Appendix** A.2 for more clarity.

## 4 Experiment

### 4.1 Experimental Settings

**Dataset**. There are three real-world datasets applied for evaluation, *i.e.,* Yelp, Amazon Fashion and Amazon Beauty. We follow the previous SRS works [18, 50] for preprocessing and data split. More details about the datasets and preprocessing can be seen in **Appendix** B.1.

**Baselines**. To validate the flexibility, we combine the competing baselines and LLM-ESR with three well-known backbone SRS models: GRU4Rec [14], Bert4Rec [49] and SASRec [18]. Then, two groups of baselines are compared in the experiments. One group is the traditional enhancement framework for the long-tailed sequential recommendation, including CITIES [17] and MELT [20]. The other group is the LLM-based enhancement framework, which contains RLMRec [45] and LLMInit [13, 16]. The more details about baselines are put into **Appendix** B.2.

**Implementation Details**. The hardware used in all experiments is an Intel Xeon Gold 6133 platform with Tesla V100 32G GPUs, while the basic software requirements are Python 3.9.5 and PyTorch 1.12.0. The hyper-parameters $N$ and $\alpha$ are searched from $\{2, 6, 10, 14, 18\}$ and $\{1, 0.5, 0.1, 0.05, 0.01\}$. More details about the implementation details are in **Appendix** B.3. The implementation code is available at https://github.com/Applied-Machine-Learning-Lab/LLM-ESR.

**Evaluation Metrics**. In the experiments, we adopt the metrics of *Top-10* list for evaluation. Specifically, the *Hit Rate* (**H@10**) and *Normalized Discounted Cumulative Gain* (**N@10**) are used. Following [18], we randomly sample 100 items that the user has not interacted with as the negatives paired

Table 1: The overall results of competing baselines and our LLM-ESR. The boldface refers to the highest score and the underline indicates the next best result of the models. "**✳**" indicates the statistically significant improvements (*i.e.,* two-sided t-test with $p < 0.05$) over the best baseline.

| Dataset | Model | Overall | | Tail Item | | Head Item | | Tail User | | Head User | |
|---|---|---|---|---|---|---|---|---|---|---|---|
| | | H@10 | N@10 | H@10 | N@10 | H@10 | N@10 | H@10 | N@10 | H@10 | N@10 |
| **Yelp** | GRU4Rec | 0.4879 | 0.2751 | 0.0171 | 0.0059 | 0.6265 | 0.3544 | 0.4919 | 0.2777 | 0.4726 | 0.2653 |
| | - CITIES | 0.4898 | 0.2749 | 0.0134 | 0.0051 | 0.6301 | 0.3543 | 0.4936 | 0.2783 | 0.4756 | 0.2618 |
| | - MELT | 0.4985 | 0.2825 | 0.0201 | 0.0079 | 0.6393 | 0.3633 | 0.5046 | 0.2865 | 0.4750 | 0.2671 |
| | - RLMRec | 0.4886 | 0.2777 | 0.0188 | 0.0067 | 0.6269 | 0.3574 | 0.4920 | 0.2804 | 0.4756 | 0.2671 |
| | - LLMInit | 0.4872 | 0.2749 | 0.0201 | 0.0072 | 0.6246 | 0.3537 | 0.4908 | 0.2775 | 0.4732 | 0.2647 |
| | - LLM-ESR | **0.5724*** | **0.3413*** | **0.0763*** | **0.0318*** | **0.7184*** | **0.4324*** | **0.5782*** | **0.3456*** | **0.5501*** | **0.3247*** |
| | Bert4Rec | 0.5307 | 0.3035 | 0.0115 | 0.0044 | 0.6836 | 0.3916 | 0.5325 | 0.3047 | 0.5241 | 0.2988 |
| | - CITIES | 0.5249 | 0.3015 | 0.0041 | 0.0014 | 0.6783 | 0.3899 | 0.5274 | 0.3032 | 0.5155 | 0.2954 |
| | - MELT | 0.6206 | 0.3770 | 0.0429 | 0.0149 | 0.7907 | 0.4836 | 0.6210 | 0.3780 | 0.6191 | 0.3733 |
| | - RLMRec | 0.5306 | 0.3039 | 0.0104 | 0.0040 | 0.6938 | 0.3922 | 0.5351 | 0.3065 | 0.5137 | 0.2936 |
| | - LLMInit | 0.6199 | 0.3781 | 0.0874 | 0.0330 | 0.7766 | 0.4797 | 0.6204 | 0.3796 | 0.6178 | 0.3723 |
| | - LLM-ESR | **0.6623*** | **0.4222*** | **0.1227*** | **0.0500*** | **0.8212*** | **0.5318*** | **0.6637*** | **0.4247*** | **0.6571*** | **0.4127*** |
| | SASRec | 0.5940 | 0.3597 | 0.1142 | 0.0495 | 0.7353 | 0.4511 | 0.5893 | 0.3578 | 0.6122 | 0.3672 |
| | - CITIES | 0.5828 | 0.3540 | 0.1532 | 0.0700 | 0.7093 | 0.4376 | 0.5785 | 0.3511 | 0.5994 | 0.3649 |
| | - MELT | 0.6257 | 0.3791 | 0.1015 | 0.0371 | 0.7801 | 0.4799 | 0.6246 | 0.3804 | 0.6299 | 0.3744 |
| | - RLMRec | 0.5990 | 0.3623 | 0.0953 | 0.0412 | 0.7474 | 0.4568 | 0.5966 | 0.3613 | 0.6084 | 0.3658 |
| | - LLMInit | 0.6415 | 0.3997 | 0.1760 | 0.0789 | 0.7785 | 0.4941 | 0.6403 | 0.4010 | 0.6462 | 0.3948 |
| | - LLM-ESR | **0.6673*** | **0.4208*** | **0.1893*** | **0.0845*** | **0.8080*** | **0.5199*** | **0.6685*** | **0.4229*** | **0.6627*** | **0.4128*** |
| **Fashion** | GRU4Rec | 0.4798 | 0.3809 | 0.0257 | 0.0101 | 0.6606 | 0.5285 | 0.3781 | 0.2577 | 0.6118 | 0.5408 |
| | - CITIES | 0.4762 | 0.3743 | 0.0252 | 0.0103 | 0.6557 | 0.5191 | 0.3729 | 0.2501 | 0.6103 | 0.5354 |
| | - MELT | 0.4884 | 0.3975 | 0.0291 | 0.0112 | 0.6712 | 0.5513 | 0.3890 | 0.2770 | 0.6173 | 0.5538 |
| | - RLMRec | 0.4795 | 0.3808 | 0.0253 | 0.0105 | 0.6603 | 0.5282 | 0.3773 | 0.2577 | 0.6120 | 0.5405 |
| | - LLMInit | 0.4864 | 0.4095 | 0.0250 | 0.0104 | 0.6702 | 0.5684 | 0.3852 | 0.2973 | 0.6177 | 0.5550 |
| | - LLM-ESR | **0.5409*** | **0.4567*** | **0.0807*** | **0.0384*** | **0.7242*** | **0.6233*** | **0.4560*** | **0.3568*** | **0.6512*** | **0.5864*** |
| | Bert4Rec | 0.4668 | 0.3613 | 0.0142 | 0.0067 | 0.6470 | 0.5024 | 0.3500 | 0.2344 | 0.6183 | 0.5258 |
| | - CITIES | 0.4926 | 0.4090 | 0.0223 | 0.0099 | 0.6799 | 0.5679 | 0.3952 | 0.2975 | 0.6190 | 0.5535 |
| | - MELT | 0.4897 | 0.3810 | 0.0059 | 0.0019 | 0.6823 | 0.5319 | 0.3842 | 0.2514 | 0.6266 | 0.5491 |
| | - RLMRec | 0.4744 | 0.3567 | 0.0044 | 0.0015 | 0.6615 | 0.4981 | 0.3626 | 0.2268 | 0.6194 | 0.5251 |
| | - LLMInit | 0.4854 | 0.4035 | 0.0328 | 0.0161 | 0.6655 | 0.5577 | 0.3773 | 0.2846 | 0.6255 | 0.5578 |
| | - LLM-ESR | **0.5487*** | **0.4529*** | **0.0525*** | **0.0225*** | **0.7462*** | **0.6243*** | **0.4629*** | **0.3460*** | **0.6599*** | **0.5916*** |
| | SASRec | 0.4956 | 0.4429 | 0.0454 | 0.0235 | 0.6748 | 0.6099 | 0.3967 | 0.3390 | 0.6239 | 0.5777 |
| | - CITIES | 0.4923 | 0.4423 | 0.0407 | 0.0214 | 0.6721 | 0.6098 | 0.3936 | 0.3392 | 0.6203 | 0.5760 |
| | - MELT | 0.4875 | 0.4150 | 0.0368 | 0.0144 | 0.6670 | 0.5745 | 0.3792 | 0.2933 | 0.6280 | 0.5729 |
| | - RLMRec | 0.4982 | 0.4457 | 0.0410 | 0.0223 | 0.6803 | 0.6143 | 0.3990 | 0.3415 | 0.6270 | 0.5808 |
| | - LLMInit | 0.5119 | 0.4492 | 0.0596 | 0.0305 | 0.6920 | 0.6159 | 0.4184 | 0.3501 | 0.6332 | 0.5777 |
| | - LLM-ESR | **0.5619*** | **0.4743*** | **0.1095*** | **0.0520*** | **0.7420*** | **0.6424*** | **0.4811*** | **0.3769*** | **0.6668*** | **0.6005*** |
| **Beauty** | GRU4Rec | 0.3683 | 0.2276 | 0.0796 | 0.0567 | 0.4371 | 0.2683 | 0.3584 | 0.2191 | 0.4135 | 0.2663 |
| | - CITIES | 0.2456 | 0.1400 | 0.1122 | 0.0760 | 0.2774 | 0.1552 | 0.2382 | 0.1346 | 0.2795 | 0.1645 |
| | - MELT | 0.3702 | 0.2161 | 0.0009 | 0.0003 | 0.4582 | 0.2675 | 0.3637 | 0.2116 | 0.3997 | 0.2365 |
| | - RLMRec | 0.3668 | 0.2278 | 0.0780 | 0.0560 | 0.4357 | 0.2688 | 0.3576 | 0.2202 | 0.4089 | 0.2626 |
| | - LLMInit | 0.4151 | 0.2713 | 0.0896 | 0.0637 | 0.4928 | 0.3208 | 0.4059 | 0.2621 | 0.4571 | 0.3133 |
| | - LLM-ESR | **0.4917*** | **0.3140*** | **0.1547*** | **0.0801*** | **0.5721*** | **0.3698*** | **0.4851*** | **0.3079*** | **0.5220*** | **0.3420*** |
| | Bert4Rec | 0.3984 | 0.2367 | 0.0101 | 0.0038 | 0.4910 | 0.2922 | 0.3851 | 0.2272 | 0.4593 | 0.2801 |
| | - CITIES | 0.3961 | 0.2339 | 0.0023 | 0.0008 | 0.4900 | 0.2895 | 0.3832 | 0.2250 | 0.4551 | 0.2746 |
| | - MELT | 0.4716 | 0.2965 | 0.0709 | 0.0291 | 0.5671 | 0.3603 | 0.4596 | 0.2865 | 0.5263 | 0.3423 |
| | - RLMRec | 0.3977 | 0.2365 | 0.0090 | 0.0032 | 0.4903 | 0.2921 | 0.3853 | 0.2277 | 0.4539 | 0.2765 |
| | - LLMInit | 0.5029 | 0.3209 | 0.0927 | 0.0451 | 0.6007 | 0.3867 | 0.4919 | 0.3117 | 0.5530 | 0.3632 |
| | - LLM-ESR | **0.5393*** | **0.3590*** | **0.1379*** | **0.0745*** | **0.6350*** | **0.4269*** | **0.5295*** | **0.3507*** | **0.5839*** | **0.3972*** |
| | SASRec | 0.4388 | 0.3030 | 0.0870 | 0.0649 | 0.5227 | 0.3598 | 0.4270 | 0.2941 | 0.4926 | 0.3438 |
| | - CITIES | 0.2256 | 0.1413 | 0.1363 | 0.0897 | 0.2468 | 0.1536 | 0.2215 | 0.1406 | 0.2441 | 0.1444 |
| | - MELT | 0.4334 | 0.2775 | 0.0460 | 0.0172 | 0.5258 | 0.3995 | 0.4233 | 0.2673 | 0.4796 | 0.3241 |
| | - RLMRec | 0.4460 | 0.3075 | 0.0924 | 0.0658 | 0.5303 | 0.3652 | 0.4365 | 0.3016 | 0.4892 | 0.3345 |
| | - LLMInit | 0.5455 | 0.3656 | 0.1714 | 0.0965 | 0.6347 | 0.4298 | 0.5359 | 0.3592 | 0.5893 | 0.3948 |
| | - LLM-ESR | **0.5672*** | **0.3713*** | **0.2257*** | **0.1108*** | **0.6486*** | **0.4334*** | **0.5581*** | **0.3643*** | **0.6087*** | **0.4032*** |

with the ground truth for calculation of the metrics. To guarantee the robustness of the experimental results, we report the average results of the triplicate test with random seeds $\{42, 43, 44\}$.

## 4.2 Overall Performance

To validate the effectiveness and flexibility of the proposed LLM-ESR, we show the overall, tail and head performance on three datasets in Table 1. At a glance, we find that the proposed LLM-ESR can outperform all competing baselines with all SRS models across all user or item groups, which verifies the usefulness of our framework. Then, we probe more conclusions by the following analysis.

Table 2: The ablation study on the Yelp dataset with SASRec as the backbone SRS model. The boldface refers to the highest score and the underline indicates the next best result of the models.

| Model | Overall | | Tail Item | | Head Item | | Tail User | | Head User | |
|---|---|---|---|---|---|---|---|---|---|---|
| | H@10 | N@10 | H@10 | N@10 | H@10 | N@10 | H@10 | N@10 | H@10 | N@10 |
| - **LLM-ESR** | **0.6673** | **0.4208** | 0.1893 | 0.0845 | **0.8080** | **0.5199** | **0.6685** | **0.4229** | **0.6627** | **0.4128** |
| - *w/o* Co-view | 0.6320 | 0.3816 | 0.1898 | 0.0856 | 0.7621 | 0.4687 | 0.6318 | 0.3823 | 0.6325 | 0.3787 |
| - *w/o* Se-view | 0.6468 | 0.4038 | 0.1105 | 0.0460 | 0.8047 | 0.5091 | 0.6459 | 0.4043 | 0.6501 | 0.4018 |
| - *w/o* SD | 0.6572 | 0.4121 | **0.2003** | **0.0898** | 0.7911 | 0.5071 | 0.6566 | 0.4130 | 0.6574 | 0.4091 |
| - *w/o* Share | 0.6595 | 0.4158 | 0.1728 | 0.0783 | 0.8027 | 0.5152 | 0.6606 | 0.4186 | 0.6552 | 0.4055 |
| - *w/o* CA | 0.6644 | 0.4160 | 0.1850 | 0.0803 | 0.8004 | 0.5119 | 0.6652 | 0.4175 | 0.6616 | 0.4105 |
| 1-layer Adapter | 0.6108 | 0.3713 | 0.1107 | 0.0469 | 0.7580 | 0.4668 | 0.6065 | 0.3702 | 0.6269 | 0.3754 |
| Random Init | 0.6440 | 0.3984 | 0.1899 | 0.0839 | 0.7777 | 0.4910 | 0.6454 | 0.4018 | 0.6388 | 0.3853 |

**Overall Comparison**. From the results, we observe that the proposed LLM-ESR leads the overall performance under both two metrics, which indicates better-enhancing effects. LLMInit is often the secondary. This phenomenon shows that the injection of semantics from LLMs actually augments the SRS. However, RLMRec often underperforms compared with other LLM-based methods, because it is devised for collaborative filtering algorithms, incompatible with SRS. As for the traditional baselines, MELT stays ahead in most cases. The reason lies in that it addresses the long-tail user and long-tail item challenges simultaneously. By comparison, CITIES is even sometimes inferior to the backbone SRS model due to the seesaw problem, *i.e.,* drastic drops for popular items.

**Long-tail Item and User Comparison**. According to the split method illustrated in Section 2, the items are grouped into Tail Item and Head Item. From Table 1, we observe that our LLM-ESR not only achieves the best on the tail item group but also gets the first place on the head item group. Such performance comparison highlights the combination of semantics and collaborative signals by our dual-view modeling. LLMInit leads the tail group across all baselines, which suggests that semantic information can benefit long-tail items. It is worth noting that CITIES sometimes perform better for the tail group but harm those popular items, which means it has a seesaw problem. Additionally, the results illustrate that MELT, LLMInit and LLM-ESR can augment the tail user group markedly. MELT is devised to enhance tail user, but underperforms our method because of its limitations to collaborative perspective. Though LLMInit can also benefit tail users by introducing semantics, it ignores the utilization of LLMs from the user side.

**Flexibility**. Table 1 shows that the proposed framework can get the largest performance improvements on all three backbone SRS models, which indicates the flexibility of LLM-ESR. By comparison, the other baselines incline to depend on the type of SRS. The traditional method, *i.e.,* CITIES and MELT, tend to perform better for GRU4Rec, while LLMInit is more beneficial to Bert4Rec and SASRec.

### 4.3 Ablation Study

The results of the ablation study are shown in Table 2. Firstly, we remove the collaborative view or semantic view to investigate the dual-view modeling, denoted as *w/o Co-view* and *w/o Se-view*. The results show that *w/o Co-view* downgrades performance dramatically on the head group, while *w/o Se-view* harms tail items evidently. Such changes indicate the distinct specialty of collaborative and semantic information, highlighting the combination of both. *w/o SD* means dropping self-distillation, which shows performance drops for long-tail users. It suggests the effects of the proposed retrieval augmented self-distillation. The results of these three variants validate the motivation for designing each component for LLM-ESR. *w/o Share* and *w/o CA* represent using split sequence encoder and removing cross-attention. The decrease in performance of these two illustrates the effectiveness of the sharing design and sequence-level fusion. More results can be seen in **Appendix** C.1.

Furthermore, we have two designs to ease the optimization of the entire LLM-ESR framework. One is that we use dimension-reduced LLM item embeddings to initialize the collaborative embedding layer instead of random initialization. On the other hand, we propose a two-layer adapter to fill the large dimension gap between LLM embeddings and item embeddings. To illustrate the effectiveness of these two designs, we compare *1-layer Adapter* and *Random Init* variants of LLM-ESR. The results, shown in Table 2, indicate that both variants underperform the original LLM-ESR, verifying the success of our special designs.

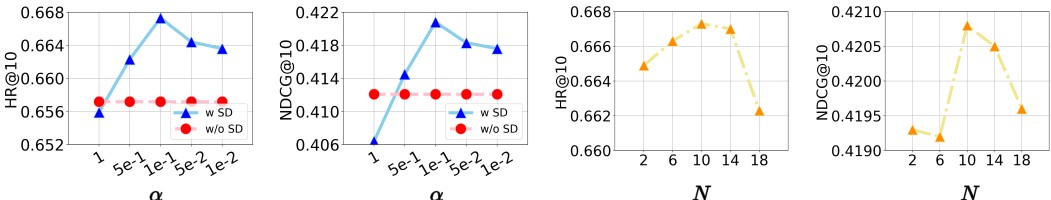

Figure 3: The hyper-parameter experiments on the weight of self-distillation loss $\alpha$ and the number of retrieved similar users $N$. The results are based on the Yelp dataset with the SASRec model.

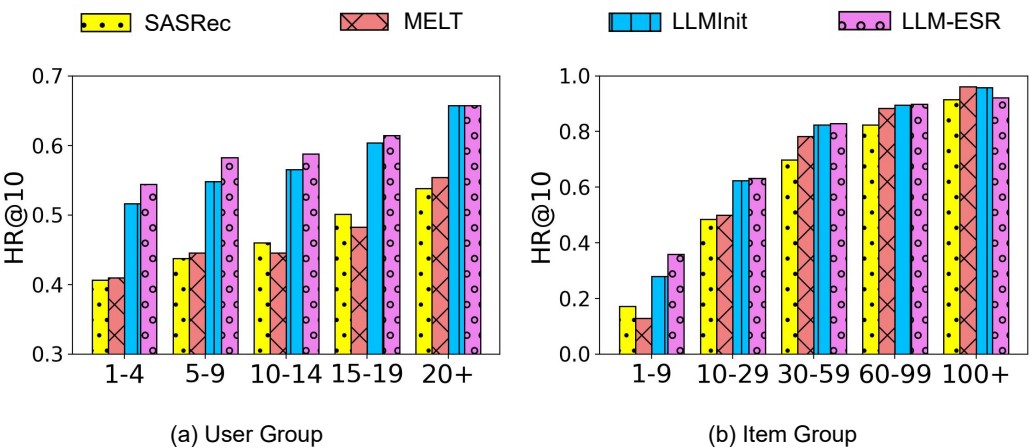

(a) User Group  (b) Item Group

Figure 4: The results of the proposed LLM-ESR and competing baselines in meticulous user and item groups. The results are based on the Beauty dataset with the SASRec model.

### 4.4 Hyper-parameter Analysis

To investigate the effects of the hyper-parameters in LLM-ESR, we show the performance trend along with their changes in Figure 3. The hyper-parameter $\alpha$ controls to what extent the designed self-distillation affects the optimization. With $\alpha$ ranging from 1 to 0.01, the recommending accuracy rises first and drops then. The reason for the compromised performance of large $\alpha$ lies in that overemphasis on self-distillation will affect the convergence of ranking loss. Smaller $\alpha$ also downgrades the performance, which indicates the usefulness of the designed self-distillation. As for the number of retrieved users $N$, the best is 10. The reason is that more users can provide more informative interactions. However, too large $N$ may decrease the relatedness of the retrieved users.

### 4.5 Group Analysis

For more meticulous analysis, we split the users and items into 5 groups according to sequence length $n_u$ and popularity $p_v$, and show the performance of each group in Figure 4. From the results, we observe that LLM-based frameworks derive increases in every user and item group, while MELT has a positive effect on some specific groups. It reflects the seesaw problem of MLET and reveals the benefit of making use of semantic embeddings from LLMs. Comparing LLMInit with LLM-ESR, LLM-ESR can get more increments on the long-tail groups (*e.g.,* 1-4 user group and 1-9 item group), which proves the better reservation of semantic information from LLMs by our framework. The group analysis of Bert4Rec and GRU4Rec as backbones are shown in **Appendix** C.3.

## 5 Related Works

### 5.1 Sequential Recommendation

The core of sequential recommendation refers to capturing the sequence pattern for the next likely item [29, 39, 31, 38, 60, 24, 23, 26, 40]. Thus, at the early stage, researchers focus on fabricating the

architecture to improve model capacity. GRU4Rec [14] and Caser [50] apply RNNs and CNNs [21] for sequence modeling. Later, inspired by the great success of self-attention [52] in natural language processing, SASRec [18] and Bert4Rec [49] verify its potential in SRS. Also, Zhou *et al.* [66] proposes a pure MLP architecture, achieving similar accuracy but higher efficiency compared with SASRec. Despite the great progress in SRS, long-tail problems are still underexplored. As for the long-tail item problem, CITIES [17] designs an embedding inference function for those long-tail items specially. In terms of the long-tail user problem, data augmentation is the main way [37, 34]. Only one work, MELT [20], addresses both two problems simultaneously but still sticks to a collaborative perspective. By comparison, the proposed LLM-ESR handles both the two long-tail problems better from a semantic view by introducing LLMs.

## 5.2 LLMs for Recommendation

Large language models [63, 43] have attracted widespread attention due to their powerful abilities in semantic understanding. Recently, There emerge several works to explore how to utilize LLMs in recommender systems (RS) [64, 28, 22, 41, 57, 58, 65, 32, 30, 35], which can be categorized into two lines, *i.e.,* LLMs as RS and LLMs enhancing RS. The first line of research aims to complete recommendation tasks by LLMs directly. At the early stage, researchers tend to fabricate the prompt templates to stimulate the recommending ability of LLMs by dialogues. For example, ChatRec [10] proposes a dialogue process to complete recommendation tasks step by step. DRDT [55] integrates a retrieval-based dynamic reflection process for SRS by in-context learning [6]. LLMRerank [9] and UniLLMRec [62] fabricate the chain-of-thought prompts to target the reranking stage and whole recommendation process, respectively. Besides, some other researchers explore fine-tuning open-sourced LLMs for RS. TALLRec [2] is the first one, which fine-tunes a LLaMA-7B by parameter-efficient fine-tuning techniques [15, 33]. Some following works, including E4SRec [25], LLaRA [27] and RecInterpreter [59], target combining collaborative signals into LLMs by modifying the tokenization. However, this line of work faces the challenge of high inference costs. Another line, LLMs enhancing RS, is more practical, because they avoid the use of LLMs while recommending. For instance, RLMRec [45] aligns with LLMs by an auxiliary loss. AlphaRec [47] adopts LLMs embedding to enhance the collaborative filtering models. On the other hand, LLM4MSR [56] and Uni-CTR [8] propose to utilize LLMs to augment the multi-domain recommendation models. As for LLMs enhancing sequential recommendation, Harte *et al.* [13] and Hu *et al.* [16] adopt LLMs embedding as the initialization for the traditional models. The proposed LLM-ESR belongs to the latter category but further alleviates the problem of defect of semantic information.

## 6 Conclusion

In this paper, we propose a large language model enhancement framework for sequential recommendation (LLM-ESR) to handle the long-tail user and long-tail item challenges. Firstly, we acquire and cache the semantic embeddings derived from LLMs, which is for inference efficiency. Then, a dual-view modeling framework is proposed to combine the semantics from LLMs and collaborative signals contained in the traditional model. It can help augment the long-tail items in SRS. Next, we design the retrieval augmented self-distillation to alleviate the long-tail user challenge. Through the comprehensive experiments, we verify the effectiveness and flexibility of our LLM-ESR.

## 7 Acknowledgements

This research was partially supported by National Key Research and Development Program of China (2022YFC3303600), National Natural Science Foundation of China (No.62192781, No.62177038, No.62293551, No.62277042, No.62137002, No.61721002, No.61937001, No.62377038), Project of China Knowledge Centre for Engineering Science and Technology, "LENOVO-XJTU" Intelligent Industry Joint Laboratory Project, Research Impact Fund (No.R1015-23), Collaborative Research Fund (No.C1043-24GF), APRC - CityU New Research Initiatives (No.9610565, Start-up Grant for New Faculty of CityU), CityU - HKIDS Early Career Research Grant (No.9360163), Hong Kong ITC Innovation and Technology Fund Midstream Research Programme for Universities Project (No.ITS/034/22MS), Hong Kong Environmental and Conservation Fund (No. 88/2022), SIRG - CityU Strategic Interdisciplinary Research Grant (No.7020046), and Tencent (CCF-Tencent Open Fund, Tencent Rhino-Bird Focused Research Program).

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

# A Supplement to Method

In this section, the details of prompt design and the procedures of LLM-ESR are addressed.

## A.1 Prompt Design

In Section 3.2 and Section 3.3, we format the attributes of items and historical interactions of users into textual prompts, for their semantic embeddings by LLMs. During the process of constructing prompts, the templates play a vital role. Here, the templates are listed as follows.

**Item Prompt Template**. The templates mainly organize the attributes and descriptions of items, which vary across distinct datasets due to different recorded attributes. In the following templates, the words underlined are the corresponding attributes that will be filled in.

> **Item Prompt Template (Yelp)**
>
> The point of interest has the following attributes:
> name is <NAME>; category is <CATEGORY>; type is <TYPE>; open status is <OPEN>;
> review count is <COUNT>; city is <CITY>; average score is <STARS>.

> **Item Prompt Template (Fashion)**
>
> The fashion item has the following attributes:
> name is <TITLE>; brand is <BRAND>; score is <DATE>; price is <PRICE>.
> The item has the following features: <FEATURE>.
> The item has the following descriptions: <DESCRIPTION>.

> **Item Prompt Template (Beauty)**
>
> The beauty item has the following attributes:
> name is <TITLE>; brand is <BRAND>; price is <PRICE>.
> The item has the following features: <CATEGORIES>.
> The item has the following descriptions: <DESCRIPTION>.

**User Prompt Template**. This template mainly organizes the items that the user has interacted with. To utilize the semantic information and avoid excess of the limitation of input length, the item in the prompt is represented by its title. Besides, the three datasets share a unique template.

> **User Prompt Template**
>
> The user has visited the following items:
> <ITEM1_TITLE>, <ITEM2_TITLE>, ...
> please conclude the user's preference.

## A.2 Train and Inference Process

For a clearer illustration of the training and inference process, we conclude them in Algorithm 1. First, the hyper-parameters and backbone SRS model are specified (lines 1-3). Then, organize the attributes of items and historical interactions into textual prompts to get their semantic embeddings (line 4). At the beginning of the training, we initialize the embedding layers in the dual-view framework (line 5). Next, calculate the ranking loss by dual-view modeling (lines 7-9) and auxiliary loss by retrieval augmented self-distillation (lines 10-11). Through the sum of these two losses (line 12), we can optimize the whole LLM-ESR. During the inference, only the dual-view modeling process is conducted to get the final recommendations (lines 16-17).

---
**Algorithm 1** Train and inference process of LLM-ESR
---
 1: Indicate the backbone sequential recommendation model $f_\theta$.
 2: Indicate the number of retrieved similar users $N$.
 3: Indicate the weight of self-distillation loss $\alpha$.
 4: Get the semantic embeddings $\mathbf{E}_{se}$ and $\mathbf{U}_{llm}$ by LLMs.

**Train Process**
 5: Initialize the embedding layers in the dual-view framework by the raw and dimension-reduced $\mathbf{E}_{se}$. Freeze the raw $\mathbf{E}_{se}$.
 6: **for** a batch of users $\mathcal{U}_B$ in $\mathcal{U}$ **do**
 7:     Get the user preference representation in semantic and collaborative views, *i.e.,* $\mathbf{u}^{se}$ and $\mathbf{u}^{co}$, respectively.
 8:     Calculate the probability score of ground-truth and negative items by Equation (5).
 9:     Calculate the ranking loss by Equation (6).
10:     Retrieve the similar users for each user in $\mathcal{U}_B$ by Equation (7).
11:     Calculate the self-distillation loss by Equation (9).
12:     Sum the ranking loss and self-distillation loss. Then, update the parameters.
13: **end for**

**Inference Process**
14: Load $\mathbf{E}_{se}$ for item embedding layers and other trained parameters.
15: **for** each user $u_k$ in $\mathcal{U}$ **do**
16:     Get the user preference representation in semantic and collaborative views, *i.e.,* $\mathbf{u}^{se}$ and $\mathbf{u}^{co}$.
17:     Calculate the probability score of each candidate item by Equation (5) and give out the final recommended list.
18: **end for**
---

# B Experimental Settings

In this section, we will refer to more details about the experimental settings.

## B.1 Dataset and Preprocessing

The comprehensive experiments in this paper are conducted on three common-used datasets, *i.e.,* Yelp, Fashion and Beauty. **Yelp**[3] is the dataset that records the check-in histories and corresponding reviews of users. We only adopt the check-in data and the attribute information of the point-of-interests. Amazon[4] [42] is a large e-commerce dataset, which includes user's reviews on commodities. There are several sub-categories in this dataset and we use two of them, *i.e.,* **Fashion** and **Beauty**.

For preprocessing, we refer to the procedures in SASRec [18]. Since the sequential recommendation is often utilized for implicit interactions, we consider all review or rate records as interactions. Then, the users with fewer than three interacted items are dropped, because we do not explore the problem of cold-start users in this paper. As for the data split, the last item $v_{n_u}$ and the penultimate item $v_{n_u-1}$ of each interaction sequence are taken out as the test and validation, respectively. The statistics of the three preprocessed datasets are shown in Table 3.

Table 3: The statistics of the preprocessed datasets

| Dataset | # Users | # Items | Sparsity | Avg.length |
|---------|---------|---------|----------|------------|
| Yelp | 15,720 | 11,383 | 99.89% | 12.23 |
| Fashion | 9,049 | 4,722 | 99.92% | 3.82 |
| Beauty | 52,204 | 57,289 | 99.99% | 7.57 |

---

[3]https://www.yelp.com/dataset
[4]https://cseweb.ucsd.edu/~jmcauley/datasets.html#amazon_reviews

## B.2 Backbone and Baseline

**Backbone Models**. To show the flexibility of our enhancement method, we test three popular sequential recommendation models in the experiments. The main distinction between these models refers to the sequence encoder $f_\theta$ and ranking loss $\mathcal{L}_{Rank}$.

- **GRU4Rec** [14]. It adopts the GRU as the sequence encoder, and sequence-to-one pairwise loss as the final ranking loss.
- **Bert4Rec** [49]. Inspired by the training pattern of Bert [19], this backbone proposes a combination between pairwise ranking loss and cloze task, which mask a proportion of items in one sequence. The sequence encoder of Bert4Rec is the stack of bi-directional self-attention layers.
- **SASRec** [18]. Compared with Bert4rec, SASRec adopts the causal self-attention layer as the basic unit of its sequence encoder. Besides, the sequence-to-sequence pairwise ranking loss is applied for optimization during the training.

There are two groups of up-to-date baselines that are compared within this paper, *i.e.,* traditional baselines and LLM-based baselines.

**Traditional Baselines**. This category split the users and items into long-tail and head groups at first. Then, they enhance the long-tail users or items by fabricated training procedures. Note that they only utilize the collaborative signals essentially and do not introduce any semantics.

- **CITIES** [17]. This work devises an embedding-inference function to refine the embeddings of long-tail items specially. Such embedding-inference function is trained by head items and used for long-tail items during inference. We follow the hyper-parameters in the original paper and code[5].
- **MELT** [20]. MELT proposes a bilateral-branch framework to enhance the long-tail users and items. One branch is trained to generate the head user representations and enhance the tail users while inference. The other branch is to recover the embeddings of head items during training and update embeddings of tail items during inference. We refer to the implementation and the hyper-parameter settings in official code[6].

**LLM-based Baselines**. The methods in this line aim to combine the semantic information derived from LLMs to enhance the recommendation models.

- **RLMRec** [45]. This baseline is one of the pioneers in utilizing the semantic embeddings derived from LLMs. However, it is designed for collaborative filtering but not sequential recommendation. For a fair comparison, we eliminate the process of profile generation during the implementation. We refer to the source code[7] of RLMRec to adapt it to sequential recommendation models.
- **LLMInit** [13, 16]. More recent works, *i.e.,* LLM2Bert4Rec [13] and SAID [16], both utilize the LLMs embedding to initialize the item embedding layer in SRS models and then fine-tune it by interaction data. In this paper, we dub this way as LLMInit.

## B.3 Implementation Details

We conduct all experiments on an Intel Xeon Gold 6133 platform with Tesla V100 32G GPUs. Besides, the implementation is based on Python 3.9.5 and PyTorch 1.12.0. In terms of the hyper-parameter search, the criterion is N@10 on the validation set. To avoid overfitting, we adopt the early stop strategy with 20-epoch patience. For the backbone SRS models, the number of GRU layers is set to 1 for GRU4Rec, while the number of self-attention layers is fixed at 2 for SASRec and Bert4Rec. Also, the dropout rate is 0.6 for Bert4Rec. In terms of the training, the batch size and learning rate are set as 128 and 0.001 for all datasets. The embedding size is 128 for all baselines, while 64 for LLM-ESR. The reason is that there are two branches in LLM-ESR, and the half size of the other unique-branch baseline is a fair setting. Then, we choose the Adam as the optimizer. The hyper-parameters $N$ and $\alpha$ for LLM-ESR are searched from $\{2, 6, 10, 14, 18\}$ and $\{1, 0.5, 0.1, 0.05, 0.01\}$. We find that the best choice is 10 for $N$ and 0.1 for $\alpha$ for all three datasets used in this paper.

---

[5]https://github.com/swonj90/CITIES
[6]https://github.com/rlqja1107/MELT
[7]https://github.com/HKUDS/RLMRec

Table 4: The ablation study on the Yelp dataset with Bert4Rec as the backbone SRS model. The boldface refers to the highest score and the underline indicates the next best result of the models.

| Model | Overall | | Tail Item | | Head Item | | Tail User | | Head User | |
|---|---|---|---|---|---|---|---|---|---|---|
| | H@10 | N@10 | H@10 | N@10 | H@10 | N@10 | H@10 | N@10 | H@10 | N@10 |
| - LLM-ESR | **0.6623** | **0.4222** | 0.1227 | 0.0500 | **0.8212** | **0.5318** | **0.6637** | **0.4247** | **0.6571** | **0.4127** |
| - *w/o* Co-view | 0.6273 | 0.3737 | 0.1272 | 0.0520 | 0.7745 | 0.4684 | 0.6296 | 0.3760 | 0.6184 | 0.3647 |
| - *w/o* Se-view | 0.6521 | 0.4125 | 0.0981 | 0.0395 | 0.8153 | 0.5224 | 0.6533 | 0.4150 | 0.6477 | 0.4031 |
| - *w/o* SD | 0.6539 | 0.4114 | **0.1299** | **0.0534** | 0.8081 | 0.5168 | 0.6539 | 0.4129 | 0.6538 | 0.4055 |
| - *w/o* Share | 0.6592 | 0.4193 | 0.1182 | 0.0480 | 0.8187 | 0.5276 | 0.6619 | 0.4229 | 0.6482 | 0.4100 |
| - *w/o* CA | 0.6368 | 0.3924 | 0.0940 | 0.0369 | 0.7966 | 0.4971 | 0.6369 | 0.3940 | 0.6367 | 0.3862 |

Table 5: The ablation study on the Yelp dataset with GRU4Rec as the backbone SRS model. The boldface refers to the highest score and the underline indicates the next best result of the models.

| Model | Overall | | Tail Item | | Head Item | | Tail User | | Head User | |
|---|---|---|---|---|---|---|---|---|---|---|
| | H@10 | N@10 | H@10 | N@10 | H@10 | N@10 | H@10 | N@10 | H@10 | N@10 |
| - LLM-ESR | **0.5724** | **0.3413** | 0.0763 | 0.0318 | **0.7184** | **0.4324** | **0.5782** | **0.3456** | 0.5501 | 0.3247 |
| - *w/o* Co-view | 0.5660 | 0.3263 | **0.0831** | **0.0331** | 0.7022 | 0.4097 | 0.5720 | 0.3310 | 0.5530 | 0.3188 |
| - *w/o* Se-view | 0.5273 | 0.3091 | 0.0441 | 0.0187 | 0.6695 | 0.3945 | 0.5316 | 0.3122 | 0.5107 | 0.2971 |
| - *w/o* SD | 0.5562 | 0.3236 | 0.0720 | 0.0309 | 0.7028 | 0.4053 | 0.5605 | 0.3371 | 0.5498 | 0.3203 |
| - *w/o* Share | 0.5661 | 0.3353 | 0.0789 | 0.0325 | 0.7124 | 0.4274 | 0.5689 | 0.3297 | **0.5536** | **0.3285** |
| - *w/o* CA | 0.5657 | 0.3327 | 0.0736 | 0.0304 | 0.7126 | 0.4128 | 0.5755 | 0.3426 | 0.5493 | 0.3227 |

Furthermore, the embeddings of LLMs are derived from the API[8] named "text-ada-embedding-002" provided by OpenAI.

# C   More Experimental Results

In this section, we will show more experimental results to further analyze the flexibility and effectiveness of our LLM-ESR.

## C.1   Ablation Study

For further analysis, we conduct the ablation study on the proposed LLM-ESR with Bert4Rec and GRU4Rec as the backbone SRS models. The results are shown in Table 4 and Table 5. At first, we probe the effects of dual-view modeling by removing one of the views, denoted as *w/o Co-view* and *w/o Se-view*. From the overall performance, these two variants both underperform, which indicates the essence of the dual-view. Besides, *w/o Co-view* downgrades the accuracy of the head item group more, while *w/o So-view* harms the long-tail item group compared with LLM-ESR. This phenomenon highlights the advantages of the collaborative view and semantic view, respectively. As for distinct SRS backbone models, we find that Bert4Rec benefits more from collaborative information, because removing the collaborative view causes a more severe performance drop. By comparison, GRU4Rec can get more enhancement from the semantic view. Then, *w/o SD* means eliminating self-distillation. It downgrades the performance of the tail user group consistently, which indicates the proposed retrieval augmented self-distillation can actually help alleviate the long-tail user challenge. *w/o Share* represents using separate sequence encoders for the dual views. This variant is a little worse than applying a shared encoder, illustrating the common pattern for both views. Another advantage of the shared encoder is higher parameter efficiency. Besides, LLM-ESR without cross-attention (*w/o CA*) is inferior to LLM-ESR totally, which indicates the effectiveness of the sequence-level fusion.

At the same time, it is risky to overfit with semantic embeddings when the textual data is scarce. To validate the robustness of our LLM-ESR, we conduct additional experiments in scenarios with limited textual data. To simulate this situation, we removed all attributes from the item descriptions except for "name" and "categories" when constructing the textual prompts for the Yelp dataset (originally using 8 attributes). This reduced the average word count of the textual prompts from 38.38 to 20.33. We used SASRec as the backbone model in these supplementary experiments, with results presented

---

[8]https://api.openai.com/v1/embeddings

Table 6: The experiments for limited text and the design of freezing semantic embedding. All the experiments are conducted on the Yelp dataset and for LLM-ESR. "Full" and "Crop" mean that we use the completed item prompt and attribute-cropped prompt to get the LLM embeddings, respectively. "w/o F" means that we train the LLM-ESR without freezing the semantic embedding layer.

| Model | Overall | | Tail Item | | Head Item | | Tail User | | Head User | |
|---|---|---|---|---|---|---|---|---|---|---|
| | H@10 | N@10 | H@10 | N@10 | H@10 | N@10 | H@10 | N@10 | H@10 | N@10 |
| Full | 0.6673 | 0.4208 | 0.1893 | 0.0845 | 0.8080 | 0.5199 | 0.6685 | 0.4229 | 0.6627 | 0.4128 |
| Full w/o F | 0.6069 | 0.3664 | 0.1284 | 0.0541 | 0.7477 | 0.4584 | 0.6028 | 0.3647 | 0.6226 | 0.3730 |
| Crop | 0.6477 | 0.4046 | 0.1478 | 0.0675 | 0.7807 | 0.4968 | 0.6468 | 0.4058 | 0.6511 | 0.3998 |
| Crop w/o F | 0.6025 | 0.3630 | 0.1247 | 0.0563 | 0.7432 | 0.4563 | 0.6004 | 0.3615 | 0.6109 | 0.3786 |

in Table 6. In the table, **Full** and **Crop** represent the use of the complete and cropped prompts, respectively. **w/o F** denotes training LLM-ESR without freezing the semantic embedding layer. The results show a decrease in performance for both Full and Crop due to the limited textual prompt. Moreover, Full w/o F and Crop w/o F yield similar results, indicating that semantic embeddings suffer from overfitting with both complete and cropped prompts. In contrast, freezing the semantic embedding layer improves performance in both scenarios and significantly benefits long-tail items, demonstrating that our design effectively alleviates the overfitting issue.

## C.2 Visualization

To further investigate how LLMs enhance the traditional SRS models, we visualize the item embeddings of SASRec, CITIES, MELT, our LLM-ESR (concatenate the semantic embedding $\mathbf{e}^{se}$ and collaborative embedding $\mathbf{e}^{co}$), and LLM using t-SNE, as shown in Figure 5. We group the items into four categories based on their popularity. The t-SNE figures reveal that the embeddings of SASRec, CITIES, and MELT tend to cluster according to item popularity. In contrast, the distribution of LLM embeddings is more uniform, indicating that the semantic relationships are not skewed by popularity. Furthermore, the embeddings of our LLM-ESR also show a more even distribution, validating that our method effectively corrects the embedding distribution in SRS and thus can enhance the performance of long-tail items.

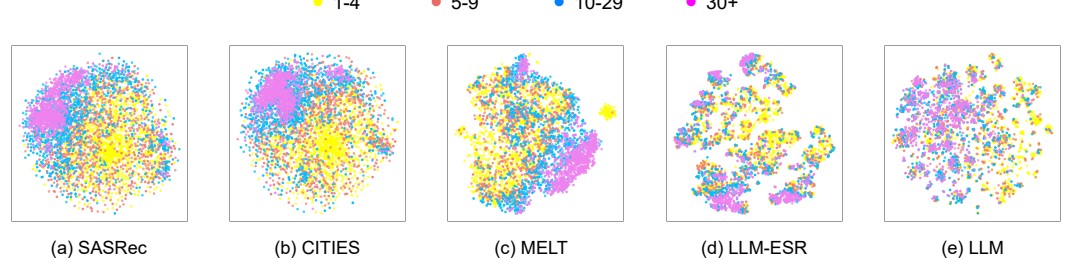

Figure 5: The visualization of the item embeddings by t-SNE. The dataset used in the experiments is Yelp. "CITIES", "MELT" and "LLM-ESR" are all based on the SASRec backbone model. "LLM" represents the embeddings derived from LLM, which encodes the semantics of textual item prompts. Different colors of circles shown in the figures mean different popularity groups of the item.

## C.3 Group Analysis

For a more meticulous analysis of to what extent the proposed LLM-ESR alleviates the long-tail challenges, we categorize users and items into 5 groups. The performances of each method with Bert4Rec and GRU4Rec as backbone models are shown in Table 6 and Tabel 7, respectively. Firstly, we analyze the results in different user groups. Undoubtedly, all methods perform worse for those users with fewer interactions, which highlights the long-tail user challenge. MELT can enhance the Bert4Rec well so that the performances in all groups get increased, but is incompatible with GRU4Rec and thus harms several groups. By comparison, LLMInit and our LLM-ESR can benefit all user groups consistently. Due to the better utilization of semantics from LLMs, LLM-ESR can outperform LLMInit evidently. Besides, the superiority is larger for more long-tailed users, *i.e.,* 1-4 and 5-9

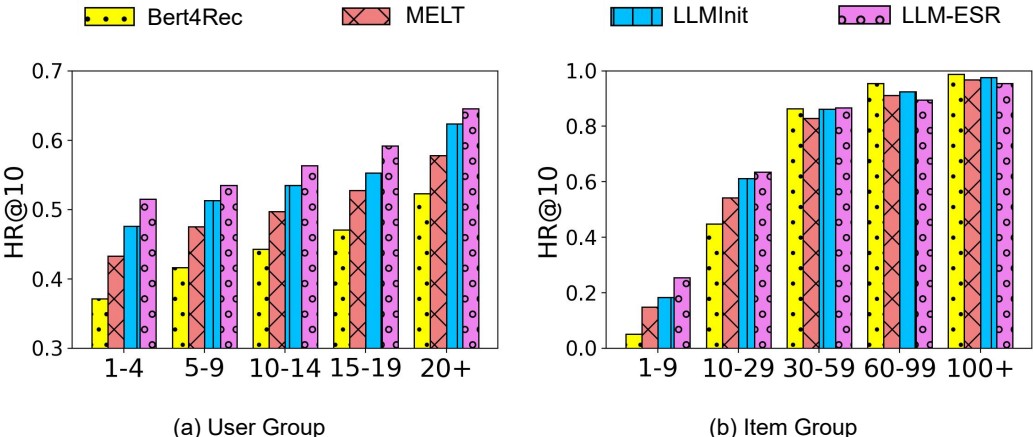

Figure 6: The results of the proposed LLM-ESR and competing baselines in meticulous user and item groups. The results are based on the Beauty dataset with the Bert4Rec model.

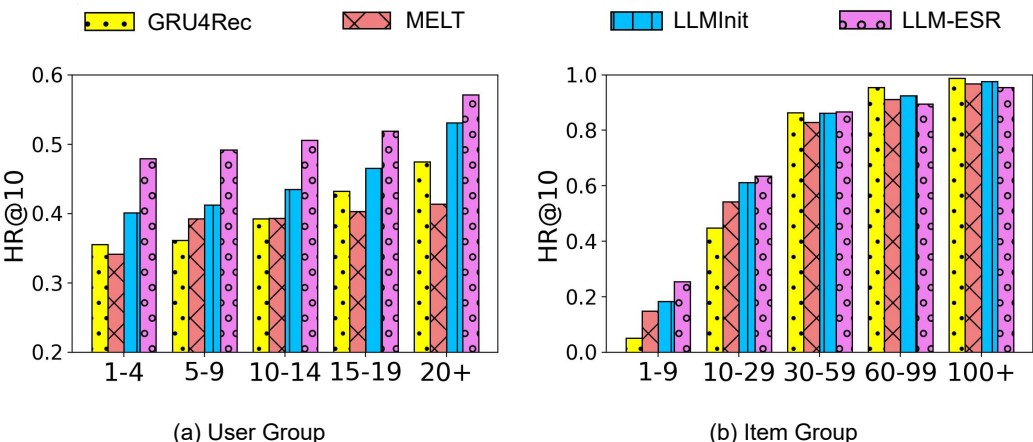

Figure 7: The results of the proposed LLM-ESR and competing baselines in meticulous user and item groups. The results are based on the Beauty dataset with the GRU4Rec model.

user groups. As for the item groups, MLET, LLMInit and LLM-ESR all elevate the recommending accuracy for long-tail items, but get a slight drop for popular items. Such a phenomenon indicates a trade-off between head and tail items. Despite that, larger increments for long-tail items of these methods result in an advance in overall performance. Also, the proposed LLM-ESR leads in 1-9 item group observably, which means it can alleviate the long-tail item challenge better.

## D  Limitation

Two potential limitations should be considered for this paper. Firstly, there are two hyper-parameters for the proposed LLM-ESR, *i.e.,* the weight of self-distillation loss $\alpha$ and the number of retrieved similar users $N$, which is time-consuming to search for the best model. Secondly, only the LLMs embedding provided by OpenAI API is validated in the experiments, but other more recent models [3, 53] may lead to better performance. Nonetheless, the experiments on various datasets and backbone models consistently validate the effectiveness of our LLM-ESR

