# OpenReview forum: "LLM-ESR: Large Language Models Enhancement for Long-tailed Sequential Recommendation"
_NeurIPS.cc/2024/Conference — NeurIPS 2024 spotlight_

### Official Review · Reviewer_Bjnq · 2024-07-08

**Soundness:** 3
**Presentation:** 3
**Contribution:** 3
**Rating:** 7
**Confidence:** 4

**Summary:**

The paper presents a framework that integrates large language models (LLMs) into sequential recommendation systems (SRS) to tackle the long-tail challenges. The framework includes dual-view modeling, which combines semantic embeddings from LLMs with collaborative signals, and a retrieval-augmented self-distillation method to enhance user preference representation. The authors validate their approach through extensive experiments on three real-world datasets, demonstrating significant improvements over existing methods.

**Strengths:**

1)	The dual-view modeling and retrieval-augmented self-distillation methods are novel contributions that enhance the performance of SRS.
2)	Utilizing LLMs to derive semantic embeddings for items and users adds a new dimension to the traditional collaborative filtering methods.
3)	The extensive experimental evaluation, including comparisons with multiple baselines and ablation studies, strengthens the validity of the findings.
4)	The paper provides comprehensive details on the methodology, including mathematical formulations and algorithmic steps, facilitating reproducibility.

**Weaknesses:**

1) There is a risk that the semantic embeddings might overfit to the training data, especially if the textual descriptions are not diverse enough.
2) The performance of the framework might be sensitive to the choice of hyper-parameters, which is not extensively explored in the paper.

**Questions:**

1) How do the authors mitigate the risk of overfitting with semantic embeddings, especially in scenarios with limited textual data?
2) Can the authors elaborate on the hyper-parameter tuning process and its impact on the performance?

**Limitations:**

The authors have addressed several limitations, but there is room for more in-depth discussion on potential biases introduced by semantic embeddings and the sensitive to the choice of hyper-parameters.

---

> ### Author Rebuttal · Authors · 2024-08-07
>
> We appreciate the reviewer's valuable time and insightful suggestions, which are important to our paper. We deliver the point-by-point response as follows.
>
> > W1 & Q1
>
> Thank you for highlighting the potential overfitting issue when textual descriptions are not diverse enough. We agree with the reviewer’s concerns. When textual prompts lack diversity, semantic embeddings may overfit the collaborative signals present in the training data. To address this, our paper proposes freezing the semantic embeddings and designing a trainable adapter during the training of LLM-ESR, which helps maintain subtle semantic differences between items.
>
> In response to the reviewer's suggestion, we conducted additional experiments in scenarios with limited textual data. To simulate this situation, we removed all attributes from the item descriptions except for "name" and "categories" when constructing the textual prompts for the Yelp dataset (originally using 8 attributes). This reduced the average word count of the textual prompts from 38.38 to 20.33. We used SASRec as the backbone model in these supplementary experiments, with results presented in __Table 2__ of the __Rebuttal PDF__.
> For convenience, we briefly show that the overall HR@10 of LLM-ESR with the original prompt changes from 0.6673 to 0.6069 when not freezing the semantic embedding layer. By comparison, the HR@10 of LLM-ESR with the limited prompt changes from 0.6477 to 0.6025.
>
> In the table, __Full__ and __Crop__ represent the use of the complete and cropped prompts, respectively. __w/o F__ denotes training LLM-ESR without freezing the semantic embedding layer. Compared with Full, the results show a decrease in performance for Crop due to the limited textual prompt. Moreover, Full w/o F and Crop w/o F yield similar results, indicating that semantic embeddings suffer from overfitting with both complete and cropped prompts. In contrast, freezing the semantic embedding layer improves performance in both scenarios and significantly benefits long-tail items, demonstrating that our design effectively alleviates the overfitting issue.
>
> We will include these experiments and analyses in the ablation study of our revised paper to further validate the effectiveness of LLM-ESR.
>
> > W2 & Q2
>
> We appreciate your concern regarding the hyper-parameters of our LLM-ESR. The key hyper-parameters are the number of similar users used for self-distillation ($N$) and the scale of the self-distillation loss ($\alpha$). We performed a grid search to optimize these hyper-parameters in the experiments. Actually, the results of this hyper-parameter tuning have been presented in __Figure 3__ of the current paper, with a detailed analysis in __Section 4.4__.
>
> The hyper-parameter $\alpha$ determines the extent to which self-distillation influences the optimization process. As $\alpha$ varies from 1 to 0.01, the recommendation accuracy initially improves and then declines. Larger values of $\alpha$ can lead to overemphasis on self-distillation, negatively impacting the convergence of the ranking loss. Conversely, smaller values of $\alpha$ reduce the effectiveness of self-distillation, highlighting its importance for our LLM-ESR.
>
> Regarding the number of retrieved users $N$, the optimal value is found to be 10. This is because a higher number of users provides more informative interactions, which helps mitigate the adverse effects of users with diverse interactions. However, if $N$ is too large, it may reduce the relevance of the retrieved users and lead to performance degradation.

---

> > ### Comment · Reviewer_Bjnq · 2024-08-12
> >
> > The authors have answered all my concerns. To this end, I prefer to recommend this paper acceptance

---

> > > ### Author Response · Authors · 2024-08-12
> > >
> > > We really thank you for taking the time to carefully assess our work and provide thoughtful feedback. Your suggestions greatly help improve our paper.

---

### Official Review · Reviewer_wsRH · 2024-07-11

**Soundness:** 4
**Presentation:** 3
**Contribution:** 3
**Rating:** 7
**Confidence:** 4

**Summary:**

This paper introduces a novel framework designed to address the long-tail challenges in sequential recommendation systems (SRS). By leveraging semantic embeddings from large language models (LLMs) and combining them with collaborative signals, the authors propose a dual-view modeling framework and a retrieval-augmented self-distillation method. This approach aims to enhance recommendations for both long-tail users and items without adding significant inference load. Extensive experiments on three real-world datasets demonstrate the effectiveness of the proposed framework.

**Strengths:**

1.	The paper successfully integrates LLMs with SRS to address long-tail challenges, a novel approach that leverages the semantic understanding of LLMs while maintaining low inference costs.
2.	The dual-view modeling framework effectively combines semantic and collaborative signals, providing a comprehensive enhancement for SRS.
3.	This method innovatively uses interactions from similar users to enhance user preference representation, addressing the long-tail user challenge.
4.	The proposed framework is model-agnostic and can be adapted to any sequential recommendation model, making it highly applicable in real-world scenarios.

**Weaknesses:**

1.	The proposed dual-view and self-distillation methods add layers of complexity to the SRS, which may pose challenges in practical implementation.
2.	The framework assumes a certain level of similarity in user interactions, which might not hold true for highly diverse user bases.
3.	Impact on Popular Items: While the focus is on long-tail items and users, the potential impact on recommendations for popular items is not thoroughly explored.

**Questions:**

1.	Could the authors provide more details on the practical implementation challenges and how they can be mitigated?
2.	How does the framework handle highly diverse user interactions where finding similar users may be challenging?
3.	Balanced Performance: What measures have been taken to ensure that the enhancement for long-tail users and items does not adversely affect recommendations for popular items?

**Limitations:**

Discussing potential negative societal impacts, such as reinforcing biases in recommendations, would be beneficial.

---

> ### Author Rebuttal · Authors · 2024-08-07
>
> We appreciate that the reviewer has raised these valuable questions to help us refine our paper. We have discussed these questions carefully and responded to them as follows.
>
> > W1 & Q1
>
> We appreciate the reviewer's advice on providing more details regarding practical implementation. There are two challenges we encountered during the implementation:
>
> 1. __Extra Inference Burden__: The dual-view modeling and self-distillation modules may increase the inference burden on SRS models. To address this challenge, we propose two efficiency solutions:
>
>    - To reduce the size of the semantic embedding layer ($|\mathcal{V}|\times d_{llm}$), we cache all the semantic embeddings transformed by the adapter before inference. Consequently, only the reduced semantic embeddings ($|\mathcal{V}|\times d$) need to be loaded during inference.
>    - The self-distillation module is eliminated during inference, as it only provides guidance for training.
>
>    With our efficient implementation, LLM-ESR introduces only a minimal additional inference burden. Originally, the model size of LLM-ESR (SASRec as the backbone) was 20.3M, and it required 12.92 seconds to test all users on the Yelp dataset. However, after applying our implementation tricks for inference, the size and time consumption reduced to 2.32M and 12.62 seconds, respectively (compared to 1.70M and 11.94 seconds for SASRec).
>
> 2. __Optimization Difficulty__: Due to the complexity of dual-view modeling and self-distillation, optimizing the entire framework can be challenging. We propose two implementation solutions to significantly alleviate this difficulty:
>
>    - The semantic and collaborative embedding layers are trained in distinct stages, resulting in optimization difficulty. Thus, we use dimension-reduced LLM item embeddings to initialize the collaborative embedding layer instead of random initialization.
>    - The dimension of LLM embeddings is much larger than the semantic embedding $\mathbf{e}^{se}$, making convergence difficult when the adapter is designed as a single linear layer. We found that a two-layer design significantly alleviates this problem. Specifically, we first reduce the dimension to half of $d_{LLM}$ and then to the item embedding size $d$.
>
>    To illustrate the effectiveness of these two implementation solutions, we compared one-layer adapter and random initialization variants of LLM-ESR in supplementary experiments. The results, shown in __Table 2__ of the __Rebuttal PDF__, indicate that both variants underperform the original LLM-ESR, verifying the success of our special designs.
>
> We will highlight these implementation details in the revised version of our paper.
>
> > W2 & Q2
>
> We appreciate the reviewer's insight into the challenge of finding similar users with highly diverse interactions.
>
> On one hand, LLMs have been shown to effectively understand user interactions through textual prompts [1,2]. Therefore, LLM embeddings of user interactions can serve as robust semantic representations of users. This suggests that embeddings generated by LLMs can assist in retrieving truly similar users.
>
> On the other hand, we acknowledge the challenge of identifying similar users due to their diverse interactions. To mitigate the adverse effects of users with diverse interactions, we average the representations of the top-N retrieved users to guide the self-distillation process. The hyper-parameter experiments shown in __Figure 3__ partially validate the reviewer's concern and the effectiveness of our design.
>
> The results indicate that LLM-ESR performs sub-optimally when N is relatively small (e.g., N=2), suggesting that inaccurate similar users can negatively impact performance. However, as N increases from 2 to 10, performance improves continuously, demonstrating that our strategy of using top-N users helps mitigate these adverse effects.
>
> Additionally, we will include a discussion on this topic in the revised paper to clarify our top-N retrieval design and its benefits.
>
> [1]. Harnessing large language models for text-rich sequential recommendation. ACM on Web Conference 2024.
>
> [2]. Tallrec: An effective and efficient tuning framework to align large language model with recommendation. 17th ACM Conference on Recommender Systems.
>
> > W3 & Q3
>
> Thank you for your advice on evaluating popular items and experienced users for our LLM-ESR.
>
> In our overall experiments (__Table 1__ in the paper), we categorize users and items into tail and head groups. Here, "Head User" and "Head Item" refer to popular items and experienced users, respectively. The results show that our LLM-ESR not only improves the performance for long-tail users and items but also enhances the performance of head users and head items compared to all baseline methods.
>
> Furthermore, for a more detailed analysis of balanced performance, we tested LLM-ESR across more granular groups in __Section 4.5__. The results in __Figure 4__ demonstrate that while LLM-ESR slightly affects the performance of extremely popular items, it benefits all other groups. These findings confirm that LLM-ESR has minimal adverse effects on popular items and experienced users.
>
> > Limitation
>
> We really appreciate your suggestion on discussing potential societal impacts. For the long-tail item issue, only recommending popular items easily causes the filter bubble problem for users, which may skew one's perspective and trap them in a small range of content. For the long-tail user issue, the ignorance of some minority groups, such as elders, may be intensified due to their less activity but it leads to more serious unfairness to the society. Thus, handling the long-tail problems in RS is essential to our society. Due to word limitations, we will add more discussions to the revised paper.

---

> > ### Author Response · Authors · 2024-08-12
> >
> > Dear reviewer, we sincerely appreciate your valuable time and insightful suggestions on our paper again. We hope that we address your concerns by our responses. Since the reviewer-author discussion deadline is approaching, please let us know if you have any other questions. We are glad to further respond to your concerns.

---

> > > ### Comment · Reviewer_wsRH · 2024-08-13
> > >
> > > You have resolved some of my concerns, and I am inclined to raise my score. However,  there are some issues that need further improvement in the final version.

---

> > > > ### Author Response · Authors · 2024-08-13
> > > >
> > > > We really thank you for your valuable time and suggestions! We promise to address the issues you have referred to in the revised version.

---

### Official Review · Reviewer_V8zP · 2024-07-11

**Soundness:** 3
**Presentation:** 3
**Contribution:** 2
**Rating:** 6
**Confidence:** 4

**Summary:**

The paper addresses the challenges in sequential recommender systems (SRS), particularly the long-tail user and long-tail item issues, which complicate user experience and seller benefits in real-world applications. The authors propose the Large Language Models Enhancement framework for Sequential Recommendation (LLM-ESR) to mitigate these challenges. The LLM-ESR framework leverages semantic embeddings derived from large language models (LLMs) to enhance SRS without increasing inference load. To tackle the long-tail item problem, the framework employs a dual-view modeling approach that integrates semantics from LLMs with collaborative signals from traditional SRS. For the long-tail user issue, a retrieval augmented self-distillation method is introduced to improve user preference representation by utilizing more informative interactions from similar users.

**Strengths:**

- The work includes extensive experiments, testing multiple aspects of the model's capabilities.

- The approach is quite new. Recommender systems based on LLMs are a promising direction.

**Weaknesses:**

- The paper does not sufficiently and deeply discuss existing work, making the motivation and core idea of the paper seem less convincing, and the innovation of the paper is also insufficient.

- The baselines used in the experiments are limited.

**Questions:**

- In line 44, the authors mention that existing studies perform poorly due to "ignorance of the true relationship between items." What is the true relationship between items, and how does it affect recommendations?

- Although SASRec is a classic model, it is not reasonable to conclude that all SRSs perform poorly in long-tail scenarios solely based on SASRec. Have the authors analyzed why SASRec performs poorly in long-tail scenarios? Do models that use other techniques specifically for long-tail scenarios have this problem? What are their limitations?

**Limitations:**

Yes

---

> ### Author Rebuttal · Authors · 2024-08-07
>
> We appreciate the meticulous and insightful comments to help us polish the paper. Please find the point-to-point responses to the reviewer's concerns.
>
> > W1 & Q1
>
> We greatly appreciate the reviewer's suggestion on refining the motivation of our paper. Existing research on long-tail issues, including CITIES [1] and MELT [2], aims to enhance the representation of long-tail items through similar popular items. However, these approaches rely on co-occurrence patterns, which overlook the semantic (expressed as "true" in the current paper) relationships between items as highlighted in our paper. Co-occurrence records inherently have an uneven distribution, often skewing item embeddings toward popularity and exacerbating the long-tail problem. In contrast, semantic relationships encoded by LLMs are unaffected by item popularity, which motivates our use of LLMs to correct the embedding distribution in SRS.
>
> For illustration, we visualized the item embeddings of SASRec, CITIES, MELT, our LLM-ESR (concatenate the semantic embedding $\mathbf{e}^{se}$ and collaborative embedding $\mathbf{e}^{co}$), and LLM using t-SNE, as shown in __Figure 1__ of the __Rebuttal PDF__. We grouped the items into four categories based on their popularity. The t-SNE figures reveal that the embeddings of SASRec, CITIES, and MELT tend to cluster according to item popularity. In contrast, the distribution of LLM embeddings is more uniform, indicating that the semantic relationships are not skewed by popularity. Furthermore, the embeddings of our LLM-ESR also show a more even distribution, validating that our method effectively corrects the embedding distribution in SRS and enhances the performance of long-tail items.
>
> We commit to including the supplementary distribution experiments and refining the illustration of our motivation in the revised paper.
>
> [1]. Jang, Seongwon, et al. Cities: Contextual inference of tail-item embeddings for sequential recommendation. 2020 IEEE International Conference on Data Mining (ICDM). IEEE, 2020.
>
> [2]. Kim, Kibum, et al. MELT: Mutual Enhancement of Long-Tailed User and Item for Sequential Recommendation. Proceedings of the 46th international ACM SIGIR conference on Research and development in information retrieval. 2023.
>
> > W2 & Q2
>
> Thank you for the advice to illustrate the limitations of existing methods more clearly.
>
> 1. Due to space limit, we present the preliminary experimental results on SASRec in the current paper. In __Figure 5__ and __Figure 6__ of __Appendix C.2__, we display the grouped performance of GRU4Rec and Bert4Rec based on item popularity and user interaction count. These results demonstrate that Bert4Rec and GRU4Rec also exhibit unsatisfactory performance for items and users with fewer interaction records, verifying that they also suffer from long-tail problems, not just SASRec.
> 2. As mentioned in our response to Q1, the main reason for long-tail problems lies in the skewed distribution of item embeddings. Additionally, to illustrate that other SRS models also have the long-tail issue, we present the item embedding distributions of GRU4Rec and Bert4Rec in __Figure 2__ of the __Rebuttal PDF__. The results indicate that the embeddings of GRU4Rec and Bert4Rec are also clustered according to item popularity. Our LLM-ESR method helps alleviate these skewed distributions, thereby enhancing the performance of long-tail items.
> 3. MELT and CITIES are two significant works designed for long-tail scenarios. We have compared their performance in __Table 1__ and conducted a more detailed group analysis of their long-tail results in __Figure 4__ of the current paper. The results demonstrate that the proposed LLM-ESR consistently outperforms these two long-tail techniques. As mentioned in response to Q1, their main limitations lie in their reliance on co-occurrence records to enhance long-tailed items and users, which did not address the skewed embedding distribution issue (as shown in __Figure 1 (b)__ and __(c)__ in the __Rebuttal PDF__).

---

> > ### Comment · Reviewer_V8zP · 2024-08-10
> >
> > Thanks for the analysis and clarification, and I will raise the rating.

---

> > > ### Author Response · Authors · 2024-08-11
> > >
> > > Thanks for your reply. Again, we really appreciate your valuable time and insightful suggestions.

---

### Author Rebuttal · Authors · 2024-08-07

We really appreciate your valuable time and insightful suggestions. The referred figures and tables of results in the rebuttal are included in the supplement PDF (i.e., __Rebuttal PDF__).

---

### Decision · Program_Chairs · 2024-09-25

**Decision:**

Accept (spotlight)

**Comment:**

Unanimous and solid reviews. Reviewers praise the experiments, approach, and relevance in real-world settings. Issues raised are mainly clarifications and seem fairly minor.